# Exposure to polystyrene nanoparticles induce disruption of mitochondrial homeostasis and impairs trophoblast cell invasion and migration via MDM2/ROCK1 pathway

Dongdong Hao☯, Tengteng Ma☯, Xiaoping Li, Fengchun Gao [ID]*

Jinan Maternity and Child Care Hospital affiliated to Shandong First Medical University, China

☯ These authors contributed equally to this work.
* Fengchungaoyx@sina.cn

## Abstract

### Background

Polystyrene nanoparticles (PS-NPs) are recognized as environmental pollutants with potential reproductive toxicity. This study delves into the impacts of PS-NPs exposure on trophoblast cells, specifically examining mitochondrial dysfunction, cell invasion and migration.

### Methods

Trophoblast cells were exposed to PS-NPs to evaluate the effects on cell proliferation, apoptosis, mitochondrial function (including mitochondrial membrane potential, intracellular ROS levels, and gene expression), autophagy, inflammatory responses and cell motility. Co-immunoprecipitation and Western blotting analyses were employed to assess the expressions and interactions of MDM2 and ROCK1 under PS-NPs exposure conditions.

### Results

We observed that PS-NPs exposure impaired trophoblast cell proliferation, promoted apoptosis, and disrupted mitochondrial function, evident by ROS elevation, mitochondrial membrane potential reduction, and altered gene expression. Increased autophagy activity and inflammatory cytokine release indicated cellular stress. Moreover, PS-NPs impeded cell migration and invasion, with exacerbated effects upon MDM2 knockdown and ROCK1 inhibition.

### Conclusion

The study elucidates the intricate connections among mitochondrial dysfunction, autophagy, inflammation, and cell motility in response to PS-NPs, suggesting that

**Data availability statement:** All relevant data are within the paper and its Supporting Information files.

**Funding:** The author(s) received no specific funding for this work.

**Competing interests:** The authors have declared that no competing interests exist.

targeting the MDM2-ROCK1 pathway could offer a promising approach to alleviate PS-NP-induced toxicity in trophoblast cells and support placental health.

## Introduction

Nanoplastics, particularly polystyrene nanoparticles (PS-NPs), have recently emerged as novel environmental pollutants, gaining significant attention owing to their small size and large specific surface area [1–3]. Nanoplastics demonstrate the ability to permeate biological barriers and accumulate in various tissues and organs [4,5]. Owing to their unique physicochemical properties, nanoplastics may pose significantly greater potential toxicity compared to that of microplastics [6]. PS-NPs, common plastic material, are widely used in products such as foam plastics, toys, CD discs, and cup lids, making them ubiquitous in the environment and increasing the potential for human and animal exposure [7–9].

Existing studies suggest PS-NPs can have detrimental effects on the reproductive system [10,11]. For example, exposure to PS-NPs has been shown to compromise oocyte quality and induce ovarian inflammation in female mice [12]. In rats, PS-NPs have been linked to ovarian fibrosis, apoptosis, and pyroptosis [13]. Furthermore, the penetration of PS-NPs into the placenta and fetal tissues through the maternal lungs affects normal fetal development in mice [14]. Therefore, investigating the reproductive toxicity of PS-NPs and their possibility to cause adverse pregnancy outcomes is of significant scientific and clinical importance. However, studies on whether PS-NPs exposure may lead to miscarriages are still limited, and their underlying mechanisms remain unclear.

Trophoblast cells, which are the primary cell type in the placenta during early to mid-pregnancy, are crucial for placenta formation and function [15]. Their invasion and migration capabilities are essential for normal placental development and, by extension, fetal oxygen and nutrient supply [16]. Previous research has indicated that PS-NPs might hinder placental development through suppressing trophoblast cell invasion, migration, and the formation of migratory vesicles [17,18]. However, the exact molecular mechanisms underlying these effects are still not completely grasped.

MDM2 (Mouse double minute 2 homolog) and ROCK1 (Rho-associated protein kinase 1) are significant regulators in the processes of cell invasion and migration and may play crucial roles in cellular dysfunction induced by PS-NPs [19–21]. MDM2, a well-known E3 ubiquitin ligase, regulates cell proliferation, apoptosis, and stress responses [22]. Beyond its role in the classical p53 pathway, MDM2 also influences cell migration, cytoskeletal stability, and mitochondrial function [23]. Through its interaction with ROCK1, MDM2 may influence cell invasion and migration, specifically by regulating ROCK1's stability or activation, which in turn affects cytoskeletal dynamics and inhibits trophoblast cell invasion and migration.

In the regulation of cell migration, morphology, and cytoskeletal reorganization, ROCK1 functions as an effector molecule downstream of Rho GTPases [24]. Through its phosphorylation of actin and non-muscle myosin, ROCK1 promotes

cell contraction and adhesion, facilitating cell motility [25]. During placental development, ROCK1 activation is critical for trophoblast cell invasion and migration [26]. Disruption of ROCK1 function can lead to developmental defects in the placental tissues. Hence, the regulation of ROCK1 activity is a crucial pathway by which PS-NPs may disrupt trophoblast cell function. Mitochondrial homeostasis is another critical aspect of cellular function. Disruption of mitochondrial integrity can result in an inadequate energy supply, increased oxidative stress, and impaired cellular processes [27]. Recent studies have shown that PS-NPs are rapidly internalized by cells, accumulating in mitochondria within minutes of exposure [28]. PS-NPs can impair mitochondrial function, leading to intracellular oxidative phosphorylation and ROS accumulation [29]. Elevated ROS levels cause oxidative damage to intracellular components, including lipids, proteins, and DNA, and can disrupt cytoskeletal dynamics, thereby impairing cellular migration [30]. Furthermore, damaged mitochondria are typically removed through autophagy, but PS-NPs may increase the mitochondrial damage load and disrupt the autophagic process, further exacerbating mitochondrial dysfunction. This study is designed to explore the potential impact of PS-NPs exposure on mitochondrial homeostasis and its influence on trophoblast cell invasion, migration, and migratory vesicle formation. By utilizing PS-NPs as a model, we aim to comprehensively evaluate their effects on trophoblast cell functionality through a series of meticulously designed experiments. Our mechanistic investigations reveal that PS-NPs disrupt mitochondrial homeostasis, trigger autophagic responses, and hinder trophoblast cell invasion, migration, and migratory vesicle formation by modulating the MDM2 and ROCK1 pathways. These novel insights shed light on the detrimental effects of inhaled PS-NPs on female reproductive health, offering valuable perspectives for further research in this area.

## Materials and methods

### Chemicals and reagents

PS-NPs employed in this research were obtained from Xuanxuan Plastic Technology Co., Ltd. (Dongguan, China). In this study, the following reagents were used: APS and TEMED (Beyotime, China); 30% acrylamide/bisacrylamide, Tris-Base, TBS buffer powder, BSA protein standard, and Tween-20(Biosharp, China); SDS, glycine, and non-fat dry milk (BioFroxx, China); 5x SDS-PAGE sample buffer and ECL chemiluminescent reagent AB (NCM Biotech, China). For RNA isolation and qPCR analysis, we used the HiScript III 1st Strand cDNA Synthesis Kit, the Cell/Tissue Total RNA Isolation Kit V2 as well as the Taq Pro Universal SYBR qPCR Master Mix. Primers used for PCR were purchased from Sangon Biotech (China). Additionally, the Prestained Protein Marker II was obtained from Servicebio (China). The primary antibodies used for Western blotting included: PINK1 polyclonal antibody, ROCK1 polyclonal antibody, NDST1 polyclonal antibody (Proteintech, China), LC3A/B (D3U4C) XP® rabbit monoclonal antibody, SQSTM1/p62 (D1Q5S) rabbit monoclonal antibody, MDM2 (D1V2Z) rabbit monoclonal antibody, Ubiquitin (P37) antibody, Anti-TSPAN4 antibody and GAPDH rabbit monoclonal antibody. The secondary antibody used for Western blotting included: Goat Anti-Rabbit IgG H&L (HRP-conjugated).

### Nanoplastic characterization

In the experiment, PS-NPs were first diluted with double-distilled water and subjected to 5 minutes of sonication. Subsequently, the samples were drop-cast onto carbon-supported copper grids for preparation and observed under transmission electron microscopy (TEM). Furthermore, we employed the Zetasizer Nano ZS to assess the hydrodynamic size distribution of the samples to determine the particle size distribution of PS-NPs sample.

### Cell culture condition

The Human Trophoblast Swan 71 cell line was obtained from Qingqi Biotechnology Development Co., Ltd. (Shanghai, China). During the experimental process, Human Trophoblast Swan 71 cells were cultured in DMEM/F12 complete medium containing 1% penicillin/streptomycin (100 µg/mL) and 10% FBS in an incubator at 37 °C with 5% $CO_2$ and 95% relative humidity.

## Experimental design

Swan 71 cells were treated under different conditions to assess the effects on trophoblast cell migration, invasion, and migratory vesicle formation. In the si-NC group, cells were transfected with a non-targeting siRNA (si-NC). In the si-NC+PS-NPs group, cells transfected with si-NC were cocultured with 200 µg/mL PS-NPs for 24 hours. The si-MDM2 group, which involved Swan 71 cells with MDM2 knockdown (si-MDM2), was not cultured with PS-NPs for 24 hours, while the si-MDM2+PS-NPs group consisted of cells with MDM2 knockdown and exposure to 200 µg/mL PS-NPs for 24 hours. In the Y-27632 group, Swan 71 cells were co-incubated with 10 µM Y-27632 for 24 hours. Lastly, the si-MDM2+PS-NPs+Y-27632 group involved cells with MDM2 knockdown, 200 µg/mL PS-NPs and 10 µM Y-27632 co-incubation for 24 hours. All experimental groups were cultured for 24 hours to evaluate the impact of the treatments on mitochondrial function and trophoblast cell behavior.

## CCK-8 assay

In this study, the proliferation capacity of Swan 71 cells was assessed by CCK-8 assay. Briefly, logarithmic-phase cells were seeded at a density of 2000 cells per well in a 96-well plate and pre-cultured for 6 hours. After adhering to the plate, the cells were subjected to different treatments according to the six groups outlined in the experiment design. After 24 hours of incubation, 10 µL of CCK-8 solution was added to each well and incubated for an additional 2 hours. Absorbance was measured at 450 nm using an ELISA reader.

### *Annexin V/PI staining*

In this study, the viability and apoptosis of cells after different treatments were observed utilizing Annexin V/PI staining in conjunction with flow cytometry analysis. Following cultivation in six different groups, cells from each group were collected for Annexin V/PI staining for 15 minutes, followed by washing with PBS. The stained cells were then subjected to analysis using flow cytometry.

## ROS detection

The intracellular levels of ROS can be accurately quantified through the utilization of DCFH-DA staining in conjunction with flow cytometry analysis. In this study, following the treatment of cells, a 10 µM working solution of DCFH-DA was applied for a duration of 30 minutes, after which the cells were carefully washed with PBS. Subsequently, the fluorescence intensity was assessed utilizing a flow cytometer, allowing for precise and quantitative evaluation.

## RT-qPCR for mtDNA and RNA expression

Mitochondrial DNA (mtDNA) and MDM2 expression was quantified using RT-qPCR (Table 1). DNA extraction was performed using a genomic DNA extraction kit. Total RNAs of human cells were extracted using the Trizol reagent. Specific primers were used for amplification of target regions, and the relative expression levels were calculated using the ΔΔCt method normalized to nuclear DNA.

**Table 1. Primer sequences of RT -qPCR.**

| Gene | Sequences (5'-3') |
|---|---|
| mtDNA | Forward: UCGUUUAGUCAUAAUAUACUG<br>Reverse: GUAUAUUAUGACUAAACGAUUAUCAUAUAAUCGUUUAGUCAU |
| MDM2 | Forward: CAGTAGCAGTGAATCTACAGGGA<br>Reverse: CAGTAGCAGTGAATCTACAGGGA |
| GAPDH | Forward: ACAACAGCCTCAAGATCATCAGC<br>Reverse: GCCATCACGCCACAGTTTCC |

### *Mitochondrial membrane potential analysis*

Tetramethyl rhodamine ethyl ester (TMRE) staining can be used for quantitative analysis of mitochondrial membrane potential (MMP). Treated cells were stained with TMRE staining solution at 37 °C for 30 minutes, washed with PBS, and then observed and quantified. The quantification involved measuring the ratio of red fluorescence to green fluorescence intensity after detection.

### Autophagy-related gene expression

The ability of Swan71 cells to express cytokines (IL-1β, TNF-α and IL-6) was assessed using an ELISA kit. Following a 12 -hours starvation treatment of the cells, different treatments were administered to each group of cells. The cell culture supernatant was collected after 24 hours of treatment. Dispense 50 μL of sample or standard into the ELISA plate and incubate at 37 °C for 30 minutes. After washing, add enzyme-conjugated antibody and continue incubating for another 30 minutes. Remove unbound antibodies, add chromogenic substrate, and incubate for 15 minutes before adding a stop solution and measuring the absorbance of each well at 450 nm.

### Western blot

In this study, the protein expression levels of migration-associated markers (TSPAN4 and NDST1), as well as MDM2 and ROCK1, were meticulously analyzed using Western blotting techniques. Cells were harvested from each experimental group and lysed using RIPA buffer that included a phosphatase inhibitor to preserve protein integrity. The protein concentrations were quantified using a BCA assay, ensuring equal loading amounts across all groups to maintain consistency in our comparative analysis. Proteins were then separated by SDS-PAGE and subsequently transferred to a PVDF membrane. The membrane was blocked with 5% non-fat milk for 30 minutes to prevent non-specific binding. This was followed by overnight incubation at 4°C with primary antibodies specific to TSPAN4 (1:2500), NDST1 (1:500), MDM2 (1:1000), and ROCK1 (1:5000). After thorough washing with TBST to remove unbound primary antibodies, the membrane was incubated with an HRP-conjugated secondary antibody for 1 hour. Following another wash to remove excess secondary antibodies, a chemiluminescent substrate was applied, and protein bands were detected using a chemiluminescence scanner. Quantitative analysis of the bands was performed using ImageJ software, providing precise measurements of protein expression levels.

### Transwell invasion and migration assay

To evaluate cell invasion and migration capabilities, Transwell assays were employed. Cells were seeded at a density of $1.5 \times 10^5$ in the upper chambers of Transwell plates using serum-free medium, while the lower chambers were filled with complete medium. After subjecting each group to different treatments, the cells were incubated for 24 hours. Post-incubation, non-migratory cells on the upper surface of the Transwell membrane were gently removed. The cells that had migrated through the pores and adhered to the lower surface were fixed with a 4% paraformaldehyde solution for 10 minutes and subsequently stained with crystal violet to facilitate visualization. Excess stain was rinsed away with PBS, and the migrated cells were examined under a microscope. This assay provides a robust measure of the cells' migratory and invasive properties under various experimental conditions.

### Co-immunoprecipitation analysis

The Co-immunoprecipitation (Co-IP) protocol was devised to explore the interaction between MDM2 and ROCK1, as well as the ubiquitination of ROCK1. Initially, cells were processed through digestion and resuspension in serum-free medium, with cell density adjusted to $1 \times 10^6$ cells per well in a 6-well plate. Following a 24-hour treatment period under experimental conditions, cell lysis was executed using RIPA buffer supplemented with PMSF, and the resulting lysates

were centrifuged, with the supernatant stored at −80°C post-collection. Protein quantification was performed using a BCA assay, and a pre-clearing step involved incubating the lysate with Protein A and Protein G agarose beads. Subsequent steps included the addition of primary antibodies and isotype control antibodies to the lysate for an overnight incubation at 4 °C, followed by immunoprecipitation, washing of the immune complexes, and resuspension of the complexes in SDS-PAGE sample buffer for analysis via SDS-PAGE and Western Blotting Analysis. The detection and quantification of MDM2, ROCK1, and ubiquitinated ROCK1 expression levels were conducted utilizing a chemiluminescence scanner for protein band visualization and ImageJ software for precise quantification, ensuring meticulous evaluation of the experimental results.

## Statistical analysis

Data are presented as mean values ± standard deviation (SD, n = 3). Statistical evaluations were performed using Graph-Pad Prism version 9 (GraphPad Inc., USA). For the analysis of differences between two distinct groups, we applied the Student's t-test. Conversely, for analyzing differences among three or more groups, one-way analysis of variance (ANOVA) was utilized. A p-value below 0.05 was deemed to indicate statistical significance, with levels of significance denoted as $*p < 0.05$, $**p < 0.01$, and $***p < 0.001$.

## Results

### Characteristics of PS-NPs

To investigate the effects and mechanisms of PS-NPs on Swan 71 cells, we first characterized the properties of PS-NPs themselves, such as morphology and particle size distribution. The transmission electron microscopy (TEM) revealed the spherical morphology of the PS-NPs (Fig 1A). Dynamic light scattering (DLS) analysis indicates that the hydrodynamic size of PS-NPs is approximate 120 nm, which is consistent with the TEM images (Fig 1B). These results confirm that PS-NPs are suitable for further cellular studies.

### Impact of PS-NPs on Swan 71 cell proliferation and apoptosis via the mitochondrial dysfunction

In this study, we first utilized siRNA to knock down MDM2 to investigate its role in PS-NPs-mediated proliferation and apoptosis of Swan 71 cells. As expected, among the three siRNA sequences, siMDM2-#1 exhibited the best knockdown efficiency and was selected for subsequent research (S1A Fig). The FAM assay also demonstrated that this transfection method effectively delivered siRNA, achieving a transfection efficiency of up to 89.9 ± 1.61% (S1B Fig). Compared

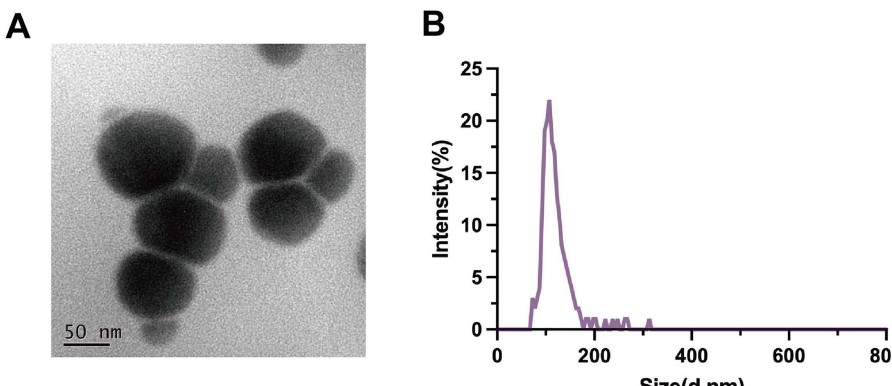

**Fig 1. Characterization of PS-NPs. (A)** Transmission electron microscopy morphology study. **(B)** DLS analysis of PS-NPs size distribution.

to the si-NC group, the proliferation capacity of cells in the PS-NPs treatment group was inhibited by approximately 40%, indicating a significant proliferative toxicity of PS-NPs on trophoblast cells. The augment in proliferation in the si-MDM2 + PS-NPs group, compared to the si-MDM2 groupsuggests that MDM2 knockdown alleviate the inhibitory effects of PS-NPs (**Fig 2A**).

In this investigation, apoptosis rates were quantitatively assessed using the Annexin V-PI assay. The results demonstrated a significant increase in apoptosis among the groups treated with PS-NPs (**Fig 2C**). Intriguingly, the induction of apoptosis in Swan 71 cells by PS-NPs appears to be mediated through the MDM2-associated pathway, a finding that was substantiated by the observation that MDM2 knockdown alleviated this apoptotic increase. Moreover, the addition of the ROCK1 inhibitor, Y-27632, further intensified the adverse effects of PS-NPs on cell viability. Specifically, in the group where cells underwent simultaneous MDM2 knockdown, PS-NPs treatment, and Y-27632 inhibition (si-MDM2 + PS-NPs + Y-27632), there was a marked reduction in cell proliferation and a notable increase in apoptosis (**Fig 2B** and **2D**). This suggests that the ROCK1 pathway plays a critical role in modulating the cellular response to PS-NP exposure, potentially exacerbating the toxic effects when inhibited in the context of MDM2 knockdown and nanoplastic treatment.

### Disruption of mitochondrial homeostasis in Swan 71 cells by PS-NPs

Following various treatments, this study utilized DCFH-DA probe for cell staining to investigate changes in intracellular ROS levels and conducted fluorescence quantification using flow cytometry. The results indicated that PS-NPs significantly elevated the intracellular ROS levels and induced oxidative stress. The increase in ROS was more pronounced in the si-NC + PS-NPs group compared to the si-MDM2 + PS-NPs group, suggesting that MDM2 depletion attenuated oxidative damage. On the contrary, ROCK1 inhibition with Y-27632 led to elevated ROS levels, supporting the notion that ROCK1 regulates oxidative stress in response to PS-NPs exposure (**Fig 3A**).

In this study, mitochondrial DNA (mtDNA) damage in different treatment groups of Swan 71 cells was further assessed using quantitative PCR (qRCR). qPCR analysis of mtDNA expression revealed a compensatory increase in PS-NPs-treated cells. However, MDM2 knockdown further decreased mtDNA levels, suggesting that MDM2 depletion enhanced the cell's ability to recover from mitochondrial damage. Inhibiting ROCK1 resulted in elevated mtDNA levels, indicating that the normal function of ROCK1 contributes to maintaining mtDNA stability in Swan 71 cells under oxidative stress conditions (**Fig 3B**). When cells are exposed to PS-NPs, the activation of MDM2 protein inhibits the activity of ROCK1, ultimately promoting ROS production, leading to cellular oxidative stress imbalance and mitochondrial homeostasis disruption, indicating that ROCK1 plays a crucial role in maintaining mitochondrial gene expression under oxidative stress conditions.

JC-1 staining was a common method used to measure cellular MMP levels. The MMP level in the si-NC + PS-NPs group decreased by approximately 50% compared to the control group. Furthermore, compared to the Si-NC + PS-NPs group, the si-MDM2 + PS-NPs group showed a slight increase in MMP levels (**Fig 3C**). Conversely, ROCK1 inhibition reduced cellular MMP levels (**Fig 3D**). This finding suggests that MDM2 knockdown and ROCK1 inhibition play opposing roles in maintaining MMP levels following PS-NPs treatment.

### Regulation of autophagy and inflammatory response in Swan 71 Cells by PS-NPs through the MDM2-ROCK1 pathway

The expression of autophagy-related proteins, including Pink1, LC3 II/LC3 I, and p62, revealed elevated autophagic activity in PS-NPs-treated cells. However, this increase was indicative of an impaired autophagic flux, likely due to mitochondrial dysfunction caused by PS-NPs. It is evident that the expression levels of multiple proteins decreased in the si-MDM2 + PS-NPs group in comparison to the si-NC + PS-NPs group. This observation suggests that the knockdown of MDM2 can effectively counterbalance the heightened autophagic activity triggered by PS-NPs. Conversely, ROCK1

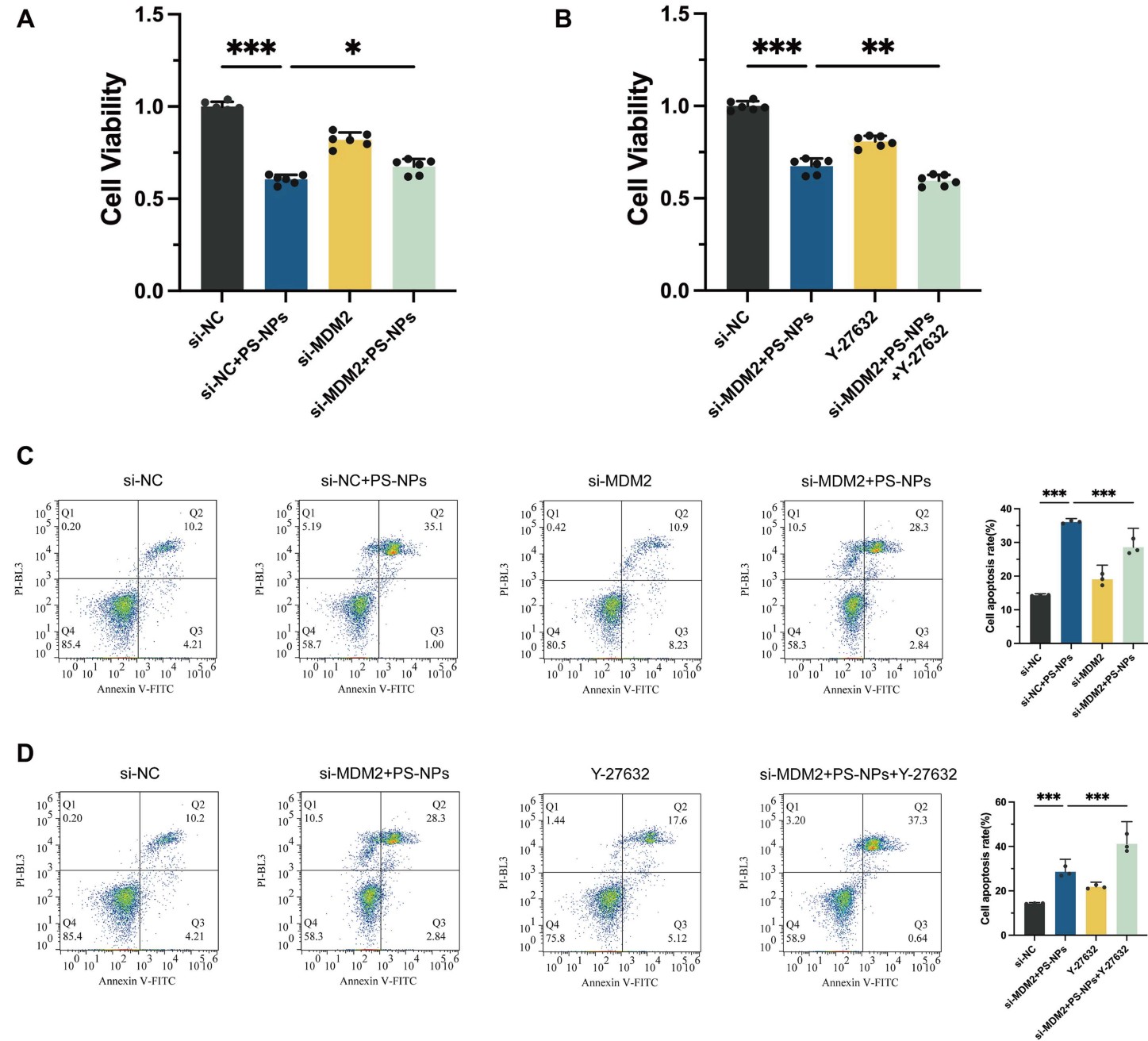

**Fig 2. Effects of PS-NPs, MDM2 knockdown, and ROCK1 inhibition on human trophoblast cell proliferation and apoptosis. (A)** CCK-8 assay was used to assess the proliferation of Swan 71 cells after PS-NPs exposure and MDM2 knockdown. **(B)** CCK-8 assay was used to assess the proliferation of Swan 71 cells after PS-NPs exposure, MDM2 knockdown and ROCK1 inhibition **(C, D)** Swan 71 cell apoptosis detection and qualification following different treatments using Annexin V/PI Staining. *$p < 0.05$, **$p < 0.01$, ***$p < 0.001$.

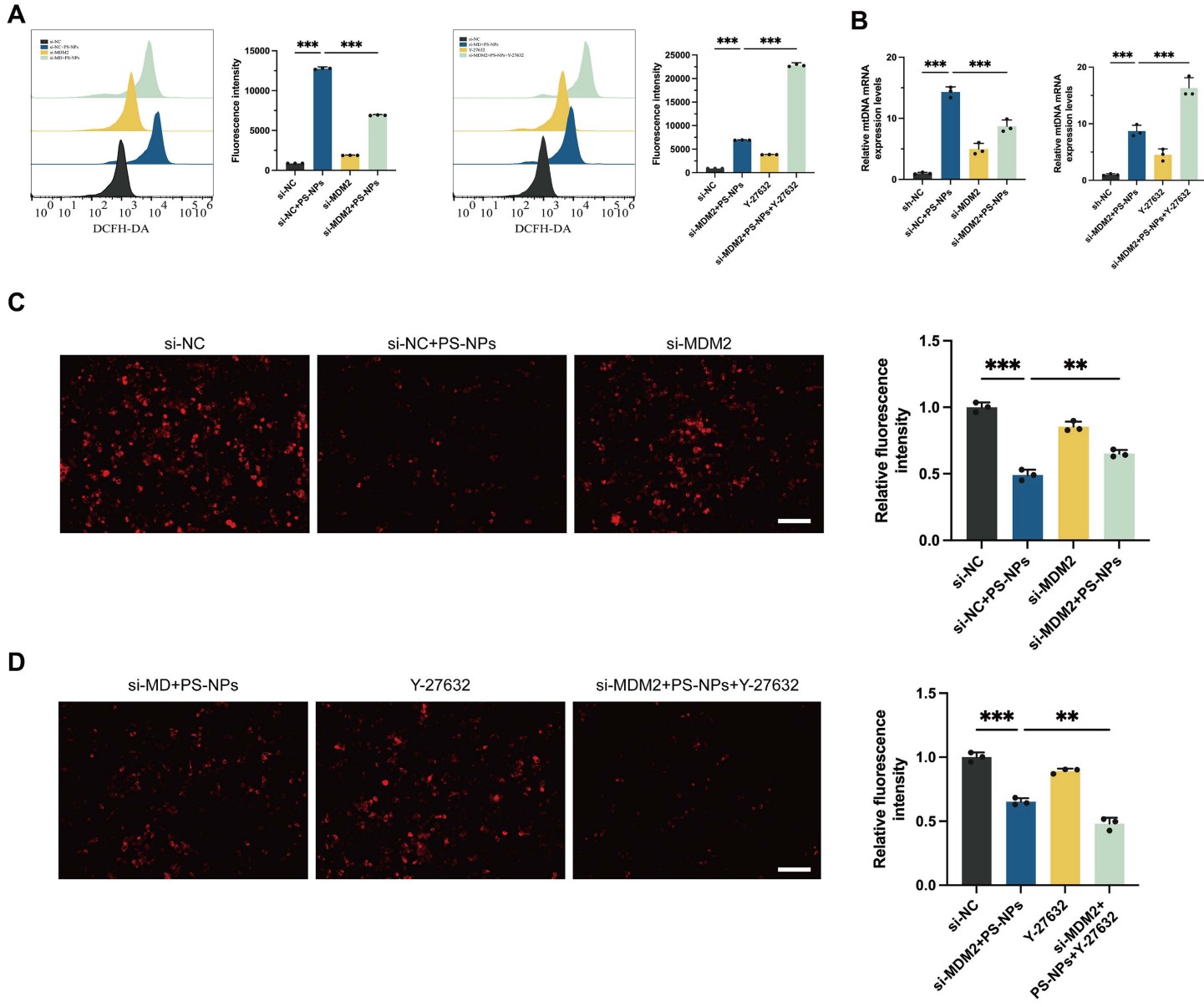

**Fig 3. Effects of PS-NPs, MDM2 knockdown and ROCK1 inhibition on disruption of mitochondrial homeostasis. (A)** Intracellular ROS levels assessed by DCFH-DA staining. Fluorescence intensity was quantified to indicate oxidative stress. **(B)** Mitochondrial DNA expression analyzed by quantitative PCR. **(C, D)** JC-1 staining results and quantitative analysis are utilized to measure the MMP levels in Swan 71 cells. Scale bar = 100 μm. *$p < 0.05$, **$p < 0.01$, ***$p < 0.001$.

inhibition exacerbates autophagy (**Fig 4A**-**4D**). This aligns with previous studies suggesting that autophagy is activated as a protective mechanism against stress-induced cellular damage.

Subsequently, this study analyzed the levels of the pro-inflammatory factors IL-1β, TNF-α and IL-6 in the cell culture supernatants of each treatment group using ELISA. Analysis of inflammatory cytokines (IL-1β, TNF-α and IL-6) revealed that their expression progressively increased in the si-NC+PS-NPs group. Further analysis revealed that compared to the group treated with PS-NPs alone, MDM2 knockdown reduced the cellular production of the three cytokines, while ROCK1

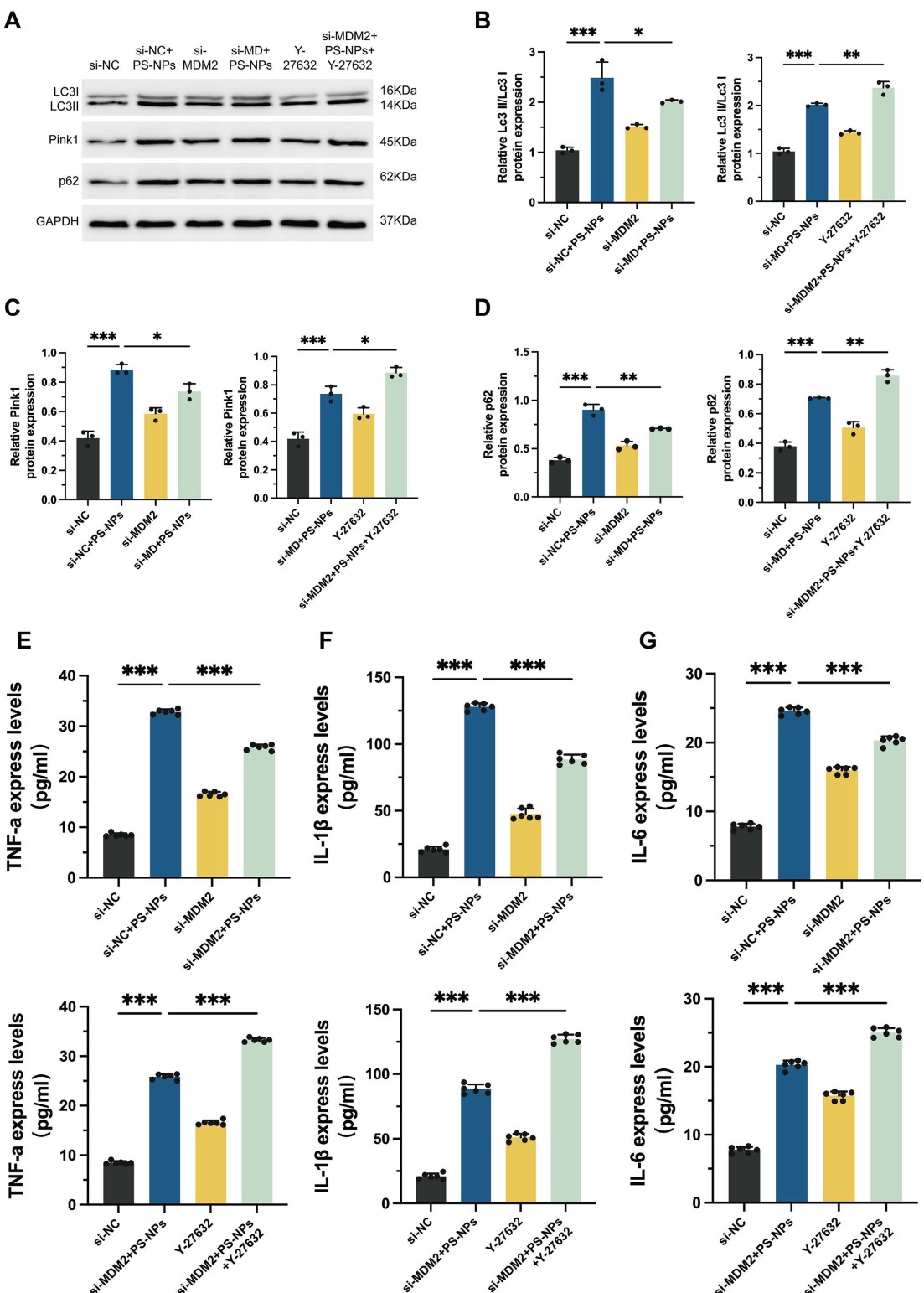

**Fig 4. Effects of PS-NPs, MDM2 knockdown and ROCK1 inhibition on autophagy and Inflammatory Response. (A)** Autophagic activity was evaluated by Western Blotting analysis of autophagy-related proteins (LC3II/LC3I, p62 and Pink1). **(B)** Quantification of LC3II/LC3I in Western Blotting analysis. **(C)** Quantification of p62 in Western Blotting analysis. **(D)** Quantification of Pink1 in Western Blotting analysis. **(E)** Expression levels of TNF-α was measured by ELISA. **(F)** Expression levels of IL-1β was measured by ELISA. **(G)** Expression levels of TNF-α was measured by ELISA. *$p < 0.05$, **$p < 0.01$, ***$p < 0.001$.

inhibition promoted the inflammatory response in Swan 71 cells (**Fig 4E**-**4G**). These findings underscore the intricate interplay among mitochondrial dysfunction, autophagy, and inflammation in the cellular response elicited by exposure to PS-NPs.

### Regulation of cell invasion and migration in Swan 71 Cells by PS-NPs via the MDM2-ROCK1 pathway

PS-NPs exposure significantly impaired trophoblast cell migration and invasion, as demonstrated by Transwell assays. Reduced expression of MDM2 alleviates the inhibition of cell migration induced by PS-NPs, suggesting the involvement of MDM2 in the impact of PS-NPs on cellular migration (**Fig 5A**). Compared to treatment with PS-NPs alone, simultaneous knockdown of MDM2 and inhibition of ROCK1 result in a more pronounced inhibition of Swan 71 cell migration, indicating that normal ROCK1 expression can mitigate the suppression of cell migration caused by PS-NPs treatment (**Fig 5B**).

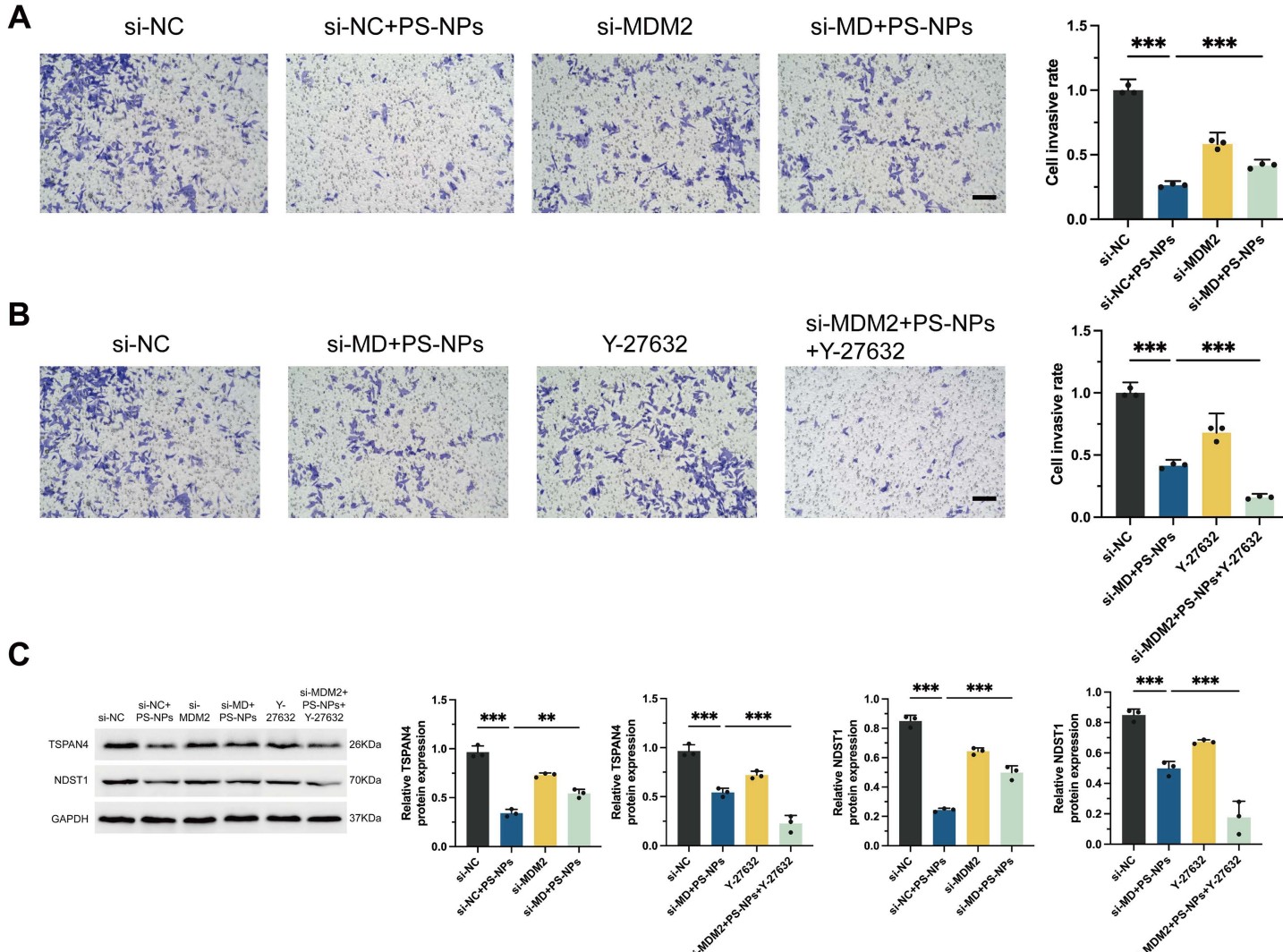

**Fig 5. Effects of PS-NPs, MDM2 knockdown, and ROCK1 inhibition on human trophoblast cells invasion and migration. (A, B)** Transwell crystal violet staining experiment is utilized to assess cell invasion and migration capabilities, along with quantitative results. Scale bar = 100 μm. **(C)** Expression of migration markers TSPAN4 and NDST1 analyzed by Western Blot analysis. $*p < 0.05$, $**p < 0.01$, $***p < 0.001$.

Western blot analysis was employed to evaluate the expression levels of migration markers TSPAN4 and NDST1, revealing significant alterations across different experimental groups. Notably, in the group exposed to PS-NPs, there was a substantial decrease in the expression of these markers: TSPAN4 expression was reduced by approximately 75%, and NDST1 expression decreased by around 80% compared to the si-NC group (Fig 5C). Across different treatments, the si-MDM2+PS-NPs+Y27632 group exhibited the lowest expression levels of TSPAN4 and NDST1 (Fig 5C). Therefore, the reduced expression of TSPAN4 and NDST1 suggests that PS-NPs impair trophoblast cell motility by disrupting the molecular pathways regulating the formation and function of these migration bodies.

### Response of MDM2 and ROCK1 to cellular damage induced by PS-NPs

To further elucidate the regulatory interactions between MDM2 and ROCK1 under PS-NPs exposure, we performed Western blotting across the experimental groups. The expression levels of MDM2 were markedly decreased in the si-MDM2 group relative to the si-NC (negative control) group (Fig 6A). This observation confirms the effective knockdown of MDM2 in the experimental setup. Importantly, upon exposure to PS-NPs, MDM2 expression was notably upregulated in the

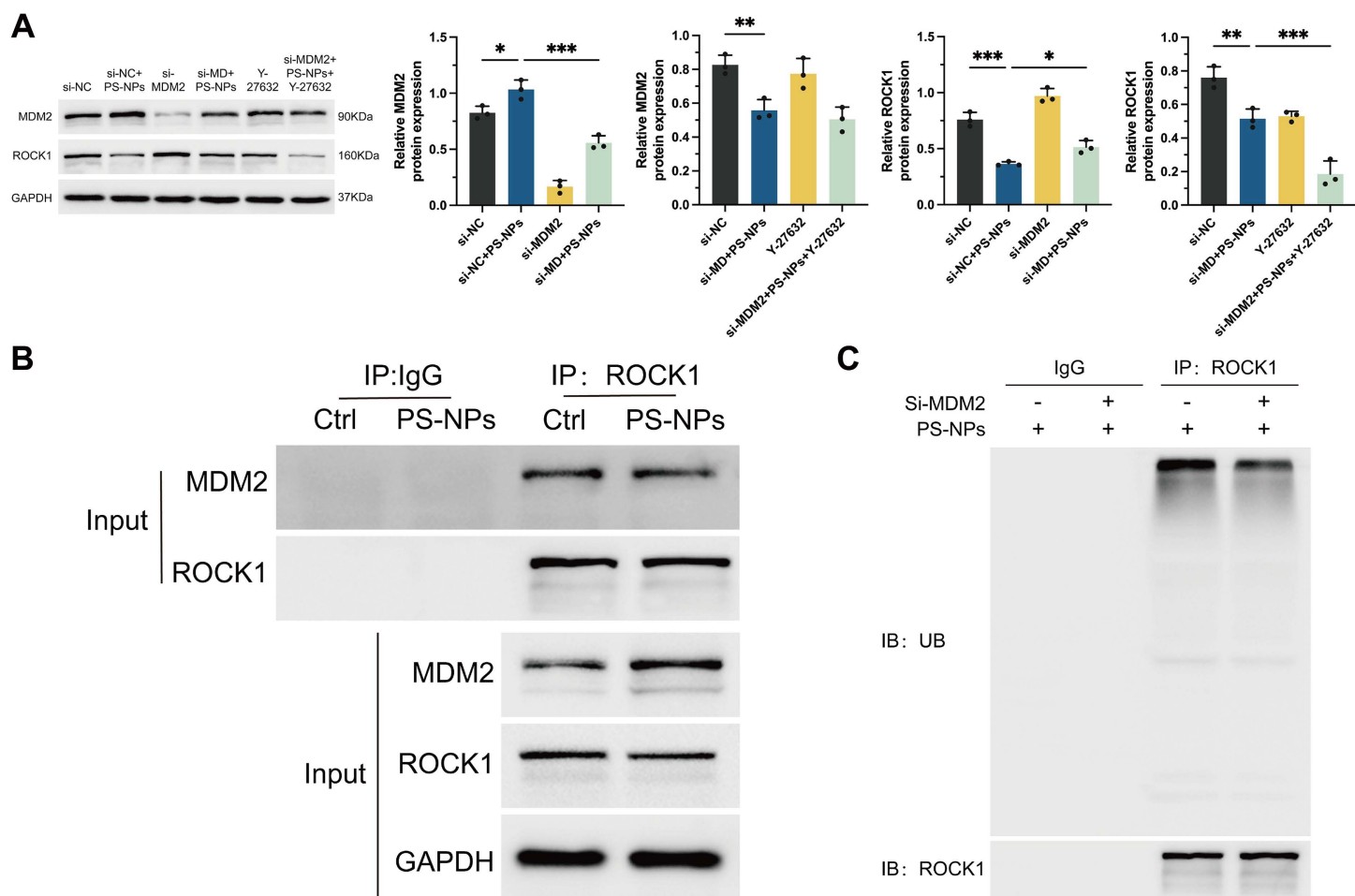

**Fig 6. Interaction between MDM2 and ROCK1 in response to PS-NPs exposure. (A)** Western Blotting analysis of MDM2 and ROCK1 expression in trophoblast cells. **(B)** Co-immunoprecipitation assay to detect the interaction between MDM2 and ROCK1 in cells exposed to PS-NPs. **(C)** Western blot analysis of ROCK1 ubiquitination. *$p < 0.05$, **$p < 0.01$, ***$p < 0.001$.

si-MDM2 + PS-NPs group, with this recovery being more pronounced compared to the si-MDM2 group. This suggests that PS-NPs exposure may induce compensatory mechanisms to restore MDM2 expression in the absence of its baseline levels. In contrast, in the Y-27632 group, MDM2 expression remained at levels comparable to the si-NC group, indicating that inhibition of ROCK1 does not affect MDM2 expression. Additionally, the si-MDM2 + PS-NPs + Y-27632 group exhibited comparable MDM2 expression levels to the si-MDM2 + PS-NPs group, indicating no significant difference between the two groups. It is suggested that ROCK1 inhibition does not modulate the PS-NPs-induced upregulation of MDM2. For ROCK1, the expression profile exhibited a distinct pattern. There was a notable increase in ROCK1 expression in the si-MDM2 group when compared to the si-NC (negative control) group. This result underscores that depletion of MDM2 leads to an upregulation of ROCK1 levels, highlighting a regulatory relationship between these two proteins.Upon PS-NPs exposure, however, ROCK1 expression was substantially reduced, with the si-MDM2 + PS-NPs group showing the most pronounced downregulation of ROCK1, suggesting that PS-NPs inhibit ROCK1 expression, potentially through oxidative stress or mitochondrial dysfunction. This differential response indicates that while MDM2 depletion increases ROCK1 expression, PS-NPs exposure overrides this effect by downregulating ROCK1 expression, likely as part of a stress response mechanism.

These results emphasize the complex interrelationship among MDM2, ROCK1, and PS-NPs in modulating cellular equilibrium. To further explore the functional relationship between MDM2 and ROCK1, we conducted Co-IP assays. The Co-IP results confirmed that MDM2 and ROCK1 interact, irrespective of PS-NPs exposure, suggesting that this interaction is a fundamental component of normal cellular physiology (Fig 6B). This interaction may contribute to the stabilization of cellular processes, and its disruption in the context of PS-NPs exposure could have significant functional consequences. An important observation from this study was the marked decrease in ROCK1 ubiquitination in the si-MDM2 + PS-NPs group relative to the si-NC + PS-NPs group The decrease in ROCK1 ubiquitination implies a pivotal function of MDM2 in governing ROCK1 stability through the ubiquitin-proteasome system. In the absence of MDM2, the degradation process of ROCK1 is compromised, resulting in an accumulation of ROCK1 protein levels. This accumulation of ROCK1 may, in turn, drive aberrant cellular signaling pathways, contributing to cellular dysfunction under PS-NPs-induced stress (Fig 6C).

## Discussion

Our research has elucidated the multifaceted impact of PS-NPs on trophoblast cellular functions, highlighting crucial insights into the molecular mechanisms underpinning cellular responses to nanoplastic exposure. Central to our findings is the role of MDM2, a critical E3 ubiquitin ligase [31], in modulating cellular responses to PS-NPs. We demonstrated that PS-NPs considerably reduce cell proliferation, potentially through mechanisms involving mitochondrial dysfunction, which are pivotal for cellular growth and viability.

Interestingly, we observed that the depletion of MDM2 mitigates the pro-apoptotic effects of PS-NPs in Swan 71 cells. This suggests that regulatory function of MDM2 on p53 stability and activity is crucial in mediating cellular resistance to PS-NPs-induced toxicity [32]. This finding is further supported by our results showing increased apoptosis rates in PS-NP-treated cells, which are exacerbated by the simultaneous inhibition of ROCK1, a regulator of cellular cytoskeletal dynamics. The inhibition of ROCK1 not only reduces cellular viability but also enhances oxidative stress, as evidenced by elevated intracellular ROS levels, thereby amplifying the detrimental effects of PS-NPs. Since ROCK1 is a key regulator of cytoskeletal dynamics and cell migration, its inhibition compromises the cell's ability to maintain viability under stress [33], further amplifying the detrimental effects of PS-NPs. Furthermore, our experiments revealed that ROCK1 inhibition disrupts MMP, a key indicator of mitochondrial health. This disruption is particularly pronounced when combined with MDM2 depletion, underscoring a complex interplay between these proteins in maintaining mitochondrial integrity under nanoplastic-induced stress.

Further highlighting the cellular adaptability to stress, our study explored the compensatory mechanisms activated in response to PS-NPs exposure. We noted that MDM2 knockdown reduces the expression of cytokines, suggesting a

modulatory role of MDM2 in the inflammatory response to PS-NPs. Conversely, ROCK1 inhibition appears to exacerbate this response, indicating its involvement in cellular stress pathways. Additionally, our findings concerning the expression of migration-related markers, TSPAN4 and NDST1, suggest that PS-NPs impair trophoblast cell motility by disrupting the molecular pathways that regulate migration bodies crucial for cell adhesion and invasion. TSPAN4 and NDST1 are specific markers of migration bodies, and their expression levels are closely associated with the presence and function of these structures [34, 35]. Migration bodies are critical for cell motility, facilitating cell adhesion, protrusion, and invasion. This impairment could have significant implications for placental function and fetal health.

The differential regulation of ROCK1 by MDM2, particularly under stress conditions induced by PS-NPs, high-lights a critical balance between these proteins in maintaining cellular homeostasis. The decreased ubiquitination of ROCK1, observed in the absence of MDM2, indicates a pivotal role for MDM2 in controlling ROCK1 stability through the ubiquitin-proteasome system. This regulatory mechanism is essential for mitigating maladaptive cellular responses to PS-NP-induced stress, which could include altered cytoskeletal dynamics and impaired cellular migration and invasion. In summary, this study provides compelling evidence of the intricate molecular interactions governing cellular responses to PS-NPs, highlighting the potential for targeted interventions that could mitigate the adverse effects of nanoplastics. While our study provides significant insights into the effects of PS-NPs on trophoblast cells, it is important to acknowledge certain limitation that could be addressed in future research. The in vitro nature of our experiments limits the extrapolation of these findings to in vivo scenarios, where the interaction between cells and nanoplastics might be influenced by a complex biological environment. Future studies could expand on this work by using animal models to explore the physiological relevance of our findings. These efforts would provide a more comprehensive understanding of the impact of nanoplastics on placental function and fetal health.

## Conclusion

Our findings suggested that PS-NPs have disrupted trophoblast cell invasion and migration through a mechanism involving mitochondrial dysfunction, autophagy dysregulation, and inflammation. Specifically, treatment of Swan71 cells with PS-NPs activated the expression of MDM2, increased ROCK1 ubiquitination, and degradation, thereby inhibiting cellular activities and functions. These results underscored the potential of targeting the MDM2-ROCK1 pathway as a therapeutic strategy to mitigate the adverse effects of PS-NPs on trophoblast cell function and, by extension, placental health.

## Supporting information

**S1 File. Raw images.**
(PDF)

**S1 Fig. Efficiency of MDM2 Knockdown by siRNA.** (A) RT-qPCR analysis of the knockdown efficiency of MDM2 using three different siRNAs. (B) Fluorescence microscopy images of Swan71 cells transfected with FAM-labeled siRNA. Scale bar = 200 μm. ***$p < 0.001$.
(TIF)

## Author contributions

**Conceptualization:** Dongdong Hao, Fengchun Gao.

**Data curation:** Dongdong Hao, Tengteng Ma, Fengchun Gao.

**Formal analysis:** Dongdong Hao, Tengteng Ma, Xiaoping Li.

**Investigation:** Tengteng Ma.

**Methodology:** Fengchun Gao.

**Resources:** Fengchun Gao.

**Supervision:** Xiaoping Li.

**Validation:** Xiaoping Li.

**Visualization:** Xiaoping Li.

**Writing – original draft:** Dongdong Hao, Tengteng Ma, Fengchun Gao.

**Writing – review & editing:** Dongdong Hao, Tengteng Ma, Xiaoping Li, Fengchun Gao.

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
