## [Decision Letter · Decision Letter 0]

23 Jun 2025

Dear Dr. Gao,

We look forward to receiving your revised manuscript.

Kind regards,

Ajit Prakash, PhD

Academic Editor

PLOS ONE

Journal Requirements:

4. Please include a caption for figure 6.

5. Please include captions for your Supporting Information files at the end of your manuscript, and update any in-text citations to match accordingly. Please see our Supporting Information guidelines for more information: http://journals.plos.org/plosone/s/supporting-information .

Reviewers' comments:

Reviewer's Responses to Questions

**Comments to the Author**

1. Is the manuscript technically sound, and do the data support the conclusions?

Reviewer #1: Yes

Reviewer #2: Yes

2. Has the statistical analysis been performed appropriately and rigorously?

Reviewer #1: Yes

Reviewer #2: Yes

3. Have the authors made all data underlying the findings in their manuscript fully available?

Reviewer #1: No

Reviewer #2: Yes

4. Is the manuscript presented in an intelligible fashion and written in standard English?

Reviewer #1: Yes

Reviewer #2: Yes

Reviewer #1: In this manuscript by Hao and Ma et al., the authors investigated the effects of Polystyrene Nanoparticles (PS-NPs) on trophoblasts. They found that when treated with PS-NPs, trophoblasts showed reduced cell proliferation and increased apoptosis. This was also accompanied by an increased cellular ROS production and decreased mitochondrial potential, indicative of mitochondrial dysfunction in these cells. Moreover, they discovered an important regulatory role of MDM2 in regulating ROCK1 levels, which are essential for the invasion and migration of trophoblasts. This research advances our mechanistic understanding in the field; however, some issues and concerns should be addressed before considering this manuscript for publication in PLOS ONE.

Major comments:

1. siRNA-mediated knockdowns (KD) may have off-target effects. Using a non-targeted siRNA is a good control, which they have already employed; however, it is generally advisable to use at least two independent siRNAs for KD experiments. Therefore, all experiments involving siRNA KD should use one additional independent siRNA for MDM2 to demonstrate a similar effect, thus eliminating any possibility of off-target effects.

Minor comments:

1. The sequences of all the qPCR primers used should be provided in a table format.

2. Catalog numbers and dilutions for all primary and secondary antibodies used should be provided.

3. Based on the error bars on the bars of siNC+PS-NPs and siMDM2+PS-NPs in Fig2 panel “C”, it does not seem to be “***” significant. Please double-check the statistical analyses and raw data and correct as needed.

4. Fig2 panel “A” is not properly referenced in the text.

5. Fig3 panels “A and B” are not properly referenced in the text.

6. Fig3 panel “A”: How can percent values (for bar graphs showing the cell apoptosis rate) be greater than 100? Please explain.

7. Fig4 is not properly referenced in the text.

8. Fig5 panel “A” is not properly referenced in the text.

9. Fig6 panels “B and C” are not properly referenced in the text.

10. The numbering of figures in the figure legends should be corrected.

11. As per PLOS data policy, please submit all the data points behind means, medians, and variance measures for all the bar graphs.

Reviewer #2: The study effectively investigates the impact of PS-NPs on various aspects of cell function. Some of the minor revisions that could be helpful are as follows:

• Some sections of the introduction have not been referenced adequately. It would be great if the authors could double-check on that.

• In the Results section, under the subheading ‘Impact of PS-NPs on Swan 21 cell proliferation and apoptosis..’, Figure 2A and 2B is mis labeled as Figure 1A and 1B. Mislabeling observed for Figure 3A as well.

• The authors could discuss the potential limitations for the study conducted and delve further into future studies that could be based on the findings from the manuscript.

**Do you want your identity to be public for this peer review?** For information about this choice, including consent withdrawal, please see our Privacy Policy

Reviewer #1: No

Reviewer #2: **Yes: ** Shikha Grover

---

## [Author Response · Author response to Decision Letter 1]

14 Aug 2025

Dear editor,

Thank you for your Email on 23-June-2025. We truly appreciate the constructive comments and suggestions from you to our manuscript (Submission ID: PONE-D-25-20220). Based on the opinions, we revised our manuscript and hope the modifications meet your expectations. The changes are in the revised manuscript. The details of our responses are as follows.

Editor comments:

1.Please ensure that your manuscript meets PLOS ONE's style requirements, including those for file naming. The PLOS ONE style templates can be found at: https://journals.plos.org/plosone/s/file?id=wjVg/PLOSOne_formatting_sample_main_body.pdf and https://journals.plos.org/plosone/s/file?id=ba62/PLOSOne_formatting_sample_title_authors_affiliations.pdf

Response: Thank you for your reminder. We have revised the manuscript as request.

Response: Thank you for your reminder. We have revised the statement as request.

Response: Thank you for your reminder. The corresponding author has provided the ORCID as request.

4. Please include a caption for figure 6.

Response: Thank you for your reminder. We have reviewed all figure captions to ensure that the caption for Figure 6 is included in the main manuscript

Response: Thank you for your reminder. We have added captions and legends for all figures in the main text.

Response: Thank you for your reminder. We have organized the original data for all the gel images and submitted it as "S1_raw_images".

Response: Thank you for your kind reminding. We have indicated in the cover letter that the uncropped original images are provided in the supporting information.

Reviewers' comments:

Reviewer #1:

In this manuscript by Hao and Ma et al., the authors investigated the effects of Polystyrene Nanoparticles (PS-NPs) on trophoblasts. They found that when treated with PS-NPs, trophoblasts showed reduced cell proliferation and increased apoptosis. This was also accompanied by an increased cellular ROS production and decreased mitochondrial potential, indicative of mitochondrial dysfunction in these cells. Moreover, they discovered an important regulatory role of MDM2 in regulating ROCK1 levels, which are essential for the invasion and migration of trophoblasts. This research advances our mechanistic understanding in the field; however, some issues and concerns should be addressed before considering this manuscript for publication in PLOS ONE.

Major comments:

1. siRNA-mediated knockdowns (KD) may have off-target effects. Using a non-targeted siRNA is a good control, which they have already employed; however, it is generally advisable to use at least two independent siRNAs for KD experiments. Therefore, all experiments involving siRNA KD should use one additional independent siRNA for MDM2 to demonstrate a similar effect, thus eliminating any possibility of off-target effects.

Response: We sincerely appreciate your professional suggestions. Before finalizing the siRNA used in our study, we initially designed three siRNAs to knock down MDM2 expression. From these, we selected siRNA-#1, which achieved a knockdown efficiency of 90%, for subsequent research. Currently, some studies also employ a single siRNA knockdown approach. Additionally, we used a FAM assay to verify the transfection efficiency of this siRNA transfection method. We have incorporated these data into the main text and the supporting information.

Page 13, line 15-20: In this study, we first utilized siRNA to knock down MDM2 to investigate its role in PS-NPs-mediated proliferation and apoptosis of Swan 71 cells. As expected, among the three siRNA sequences, siMDM2-#1 exhibited the best knockdown efficiency and was selected for subsequent research (Figure S1A).The FAM assay also demonstrated that this transfection method effectively delivered siRNA, achieving a transfection efficiency of up to 89.9 ± 1.61% (Figure S1B).

Figure S1. Efficiency of MDM2 Knockdown by siRNA. (A) RT-qPCR analysis of the knockdown efficiency of MDM2 using three different siRNAs. (B) Fluorescence microscopy images of Swan71 cells transfected with FAM-labeled siRNA.

Minor comments:

1. The sequences of all the qPCR primers used should be provided in a table format.

Response: Thank you for your suggestion. We have added the RT-qPCR primer information to Table 1.

Page 9, line 20:

Table 1. Primer sequences of RT -qPCR

Primer Forward primer Reverse primer

mtDNA UCGUUUAGUCAUAAUAUACUG GUAUAUUAUGACUAAACGAUUAUCAUAUAAUCGUUUAGUCAU

MDM2 CAGTAGCAGTGAATCTACAGGGA CAGTAGCAGTGAATCTACAGGGA

GAPDH ACAACAGCCTCAAGATCATCAGC GCCATCACGCCACAGTTTCC

2. Catalog numbers and dilutions for all primary and secondary antibodies used should be provided.

Response: Thank you very much for your suggestion. We have included the information regarding the antibody dilution factor in the revised manuscript. The dilution factors are provided in parentheses.

Page 11, line 7-8�This was followed by overnight incubation at 4°C with primary antibodies specific to TSPAN4 (1:2500), NDST1 (1:500), MDM2 (1:1000), and ROCK1 (1:5000).

3. Based on the error bars on the bars of siNC+PS-NPs and siMDM2+PS-NPs in Fig2 panel “C”, it does not seem to be “***” significant. Please double-check the statistical analyses and raw data and correct as needed.

Response: Thank you for your reminder. We have re-verified the accuracy of our statistical data. While the significance of the data differences may not be immediately apparent from the bar graphs, calculations show that the p-value for the comparison between the two groups is less than 0.001. Below is the result calculated using GraphPad Prism software.

4. Fig2 panel “A” is not properly referenced in the text.

Response: Thank you very much for your suggestion. We have removed the superfluous references and now only cite them after the final sentence.

Page 14, line 1-2�The augment in proliferation in the si-MDM2 + PS-NPs group, compared to the si-MDM2 groupsuggests that MDM2 knockdown alleviate the inhibitory effects of PS-NPs (Figure 2A).

5. Fig3 panels “A and B” are not properly referenced in the text.

Response: Thank you for your reminder. We have eliminated the redundant references and now include them only after the final sentence.

Page 14, line 23- Page 15, line 2�On the contrary, ROCK1 inhibition with Y-27632 led to elevated ROS levels, supporting the notion that ROCK1 regulates oxidative stress in response to PS-NPs exposure (Figure 3A).

Page 15, line 6-8�Inhibiting ROCK1 resulted in elevated mtDNA levels, indicating that the normal function of ROCK1 contributes to maintaining mtDNA stability in Swan 71 cells under oxidative stress conditions (Figure 3B)

6. Fig3 panel “A”: How can percent values (for bar graphs showing the cell apoptosis rate) be greater than 100? Please explain.

Response: We apologize for the incorrect labeling of the Y-axis title in Figure 3A. Thank you for pointing this out. We have corrected Figure 3-A in the revised manuscript, as shown below.

Figure 3. Effects of PS-NPs, MDM2 knockdown and ROCK1 inhibition on disruption of mitochondrial homeostasis. (A) Intracellular ROS levels assessed by DCFH-DA staining. Fluorescence intensity was quantified to indicate oxidative stress. (B) Mitochondrial DNA expression analyzed by quantitative PCR. (C, D) JC-1 staining results and quantitative analysis are utilized to measure the MMP levels in Swan 71 cells. Scale bar = 100 μm. *p < 0.05, **p < 0.01, ***p < 0.001

7. Fig4 is not properly referenced in the text.

Response: Thank you for your suggestion. We have appropriately placed the references in the text.

Page 16, line 6-9�This observation suggests that the knockdown of MDM2 can effectively counterbalance the heightened autophagic activity triggered by PS-NPs. Conversely, ROCK1 inhibition exacerbates autophagy (Figure 4A-D).

Page 16, line 15-18�Further analysis revealed that compared to the group treated with PS-NPs alone, MDM2 knockdown reduced the cellular production of the three cytokines, while ROCK1 inhibition promoted the inflammatory response in Swan 71 cells (Figure 4E-4G).

8. Fig5 panel “A” is not properly referenced in the text.

Response: Thank you for your suggestion. We have appropriately placed the references in the text.

Page 16, line 24- Page 16, line 1: Reduced expression of MDM2 alleviates the inhibition of cell migration induced by PS-NPs, suggesting the involvement of MDM2 in the impact of PS-NPs on cellular migration (Figure 5A).

9. Fig6 panels “B and C” are not properly referenced in the text.

Response: Thank you for your suggestion. We have appropriately placed the references in the text.

Page 18, line 21-24: The Co-IP results confirmed that MDM2 and ROCK1 interact, irrespective of PS-NPs exposure, suggesting that this interaction is a fundamental component of normal cellular physiology (Figure 6B).

Page 19, line 8-10: This accumulation of ROCK1 may, in turn, drive aberrant cellular signaling pathways, contributing to cellular dysfunction under PS-NPs-induced stress (Figure 6C).

10. The numbering of figures in the figure legends should be corrected.

Response: Thank you for your suggestion. We have reorganized the figure numbering.

11. As per PLOS data policy, please submit all the data points behind means, medians, and variance measures for all the bar graphs.

Response: Thank you for your suggestion. Data points are indeed crucial for the accuracy of bar graphs. In accordance with PLOS data policies, we have now displayed the data points for each bar graph.

Figure 2. Effects of PS-NPs, MDM2 knockdown, and ROCK1 inhibition on human trophoblast cell proliferation and apoptosis. (A) CCK-8 assay was used to assess the proliferation of Swan 71 cells after PS-NPs exposure and MDM2 knockdown. (B) CCK-8 assay was used to assess the proliferation of Swan 71 cells after PS-NPs exposure, MDM2 knockdown and ROCK1 inhibition (C, D) Swan 71 cell apoptosis detection and qualification following different treatments using Annexin V/PI Staining. *p < 0.05, **p < 0.01, ***p < 0.001

Figure 3. Effects of PS-NPs, MDM2 knockdown and ROCK1 inhibition on disruption of mitochondrial homeostasis. (A) Intracellular ROS levels assessed by DCFH-DA staining. Fluorescence intensity was quantified to indicate oxidative stress. (B) Mitochondrial DNA expression analyzed by quantitative PCR. (C, D) JC-1 staining results and quantitative analysis are utilized to measure the MMP levels in Swan 71 cells. Scale bar = 100 μm. *p < 0.05, **p < 0.01, ***p < 0.001

Figure 4. Effects of PS-NPs, MDM2 knockdown and ROCK1 inhibition on autophagy and Inflammatory Response. (A) Autophagic activity was evaluated by Western Blotting analysis of autophagy-related proteins (LC3II/LC3I, p62 and Pink1). (B) Quantification of LC3II/LC3I in Western Blotting analysis. (C) Quantification of p62 in Western Blotting analysis. (D) Quantification of Pink1 in Western Blotting analysis. (E) Expression levels of TNF-α was measured by ELISA. (F) Expression levels of IL-1β was measured by ELISA. (G) Expression levels of TNF-α was measured by ELISA. *p < 0.05, **p < 0.01, ***p < 0.001

Figure 5. Effects of PS-NPs, MDM2 knockdown, and ROCK1 inhibition on human trophoblast cells invasion and migration. (A, B) Transwell crystal violet staining experiment is utilized to assess cell invasion and migration capabilities, along with quantitative results. Scale bar = 100 μm. (C) Expression of migration markers TSPAN4 and NDST1 analyzed by Western Blot analysis. *p < 0.05, **p < 0.01, ***p < 0.001

Figure 6. Interaction between MDM2 and ROCK1 in response to PS-NPs exposure. (A) Western Blotting analysis of MDM2 and ROCK1 expression in trophoblast cells. (B) Co-immunoprecipitation assay to detect the interaction between MDM2 and ROCK1 in cells exposed to PS-NPs. (C) Western blot analysis of ROCK1 ubiquitination. *p < 0.05, **p < 0.01, ***p < 0.001

Reviewer #2:

The study effectively investigates the impact of PS-NPs on various aspects of cell function. Some of the minor revisions that could be helpful are as follows:

• Some sections of the introduction have not been referenced adequately. It would be great if the authors could double-check on that.

Response: Thank you for your recognition and valuable comments on our research. We have reviewed the citations in the introduction section and added references to the relevant literature where appropriate.

[16] Jeong S, Fuwad A, Yoon S, et al. A Microphysiological Model to Mimic the Placental Remodeling during Early Stage of Pregnancy under Hypoxia-Induced Trophoblast Invasion [J]. Biomimetics (Basel). 2024, 9(5): 289.

[18] Lin LY, Kantha P, Horng JL. Toxic effects of polystyrene nanoparticles on the development, escape locomotion, and lateral-line sensory function of zebrafish embryos[J]. Comp Biochem Physiol C Toxicol Pharmacol. 2023, 272:109701.

[22] Chibaya L, Karim B, Zhang H, Jones SN. Mdm2 phosphorylation by Akt regulates the p53 response to oxidative stress to promote cell proliferation and tumorigenesis [J]. Proc Natl Acad Sci U S A. 2021, 118(4): e2003193118.

[25] Okumura N, Fujii K, Kagami T, et al. Activation of the Rho/Rho Kinase Signaling Pathway Is Involved in Cell Death of Corneal Endothelium[J]. Invest Ophthalmol Vis Sci. 2016, 57(15):6843-6851.

[26] Okae H, Toh H, Sato T, et al. Derivation of Human Trophoblast Stem Cells[J]. Cell Stem Cell. 2018, 22(1):50-63.

[27] Shao Y, Hu J, Yan K, et al. Impaired mitochondrial integrity and compromised energy production underscore the mechanism underlying CoASY protein-associated neurodegeneration[J]. Cell Mol Life Sci. 2025, 82(1):84.

[30] Stojkov D, Amini P, Oberson K, et al. ROS and glutathionylation balance cytoskeletal dynamics in neutrophil extracellular trap formation[J]. J Cell Biol. 2017, 216(12): 4073-4090.

• In the Results section, under the subheading ‘Impact of PS-NPs on Swan 21 cell proliferation and apoptosis..’, Figure 2A and 2B is mis labeled as Figure 1A and 1B. Mislabeling obs

---

## [Decision Letter · Decision Letter 1]

7 Nov 2025

Exposure to Polystyrene Nanoparticles Induce Disruption of Mitochondrial Homeostasis and Impairs Trophoblast Cell Invasion and Migration via MDM2/ROCK1 Pathway

PONE-D-25-20220R1

Dear Dr. Gao,

We’re pleased to inform you that your manuscript has been judged scientifically suitable for publication and will be formally accepted for publication once it meets all outstanding technical requirements.

Kind regards,

Ajit Prakash, PhD

Academic Editor

PLOS ONE

Additional Editor Comments (optional):

Reviewers' comments:

Reviewer's Responses to Questions

**Comments to the Author**

Reviewer #1: All comments have been addressed

Reviewer #2: All comments have been addressed

2. Is the manuscript technically sound, and do the data support the conclusions?

Reviewer #1: Yes

Reviewer #2: Yes

3. Has the statistical analysis been performed appropriately and rigorously?

Reviewer #1: Yes

Reviewer #2: Yes

4. Have the authors made all data underlying the findings in their manuscript fully available?

Reviewer #1: Yes

Reviewer #2: Yes

5. Is the manuscript presented in an intelligible fashion and written in standard English?

Reviewer #1: Yes

Reviewer #2: Yes

Reviewer #1: (No Response)

Reviewer #2: The authors have adequately addressed the revisions requested. The article sounds more relevant and informative for the readers.

**Do you want your identity to be public for this peer review?** For information about this choice, including consent withdrawal, please see our Privacy Policy

Reviewer #1: No

Reviewer #2: **Yes: ** Shikha Grover

---

## [Editor Report · Acceptance letter]

PONE-D-25-20220R1

PLOS ONE

Dear Dr. Gao,

I'm pleased to inform you that your manuscript has been deemed suitable for publication in PLOS ONE. Congratulations! Your manuscript is now being handed over to our production team.

Kind regards,

on behalf of

Dr. Ajit Prakash

Academic Editor

PLOS ONE